# Women’s Knowledge and Perceptions of the Effect of Exercise during Pregnancy: A Cross-Sectional Study

**DOI:** 10.3390/ijerph20031822

**Published:** 2023-01-19

**Authors:** Vilma Dudonienė, Raija Kuisma

**Affiliations:** 1Department of Health Promotion and Rehabilitation, Lithuanian Sports University, LT-44221 Kaunas, Lithuania; 2Department of Physiotherapy, Karelia University of Applied Sciences, FI-80200 Joensuu, Finland

**Keywords:** exercise, pregnancy, knowledge, perceptions

## Abstract

Background: Many women may not be aware of the effect of exercise during pregnancy. The objective of this study was to explore the knowledge and perceptions of the effect of exercise and compare whether there was a difference between women who were pregnant at the time of the study, who were not pregnant but had given birth in the past, and those who had never been pregnant and had never given birth. Methods: A cohort of 291 women, aged 18–55 years, participated in this cross-sectional study. Ninety-one (31.3%) women were pregnant at the time of the study; 97 (33.3%) were not pregnant but had given birth in the past, and 103 (35.4%) were not pregnant and had never given birth. This was a survey by an on-line questionnaire. Results: Only 24.4 % of respondents were aware of the effect of exercise in pregnancy, 44% were not aware of the effect of exercise in pregnancy, and 52.6% did not know if exercise interventions could be prescribed during pregnancy. The perceived aims of exercise, reported by the women, were to keep the body fit (58%), to reduce low back and pelvic pain (55%), and to facilitate childbirth (51%). Conclusions: Women who were not pregnant but had given birth were significantly less aware of the effect of exercise than women who were pregnant at the time of the study or not pregnant and had never given birth. The internet was the most common source of information about exercise among all respondents. Almost all women in the study felt the need for more knowledge about the effect of exercise during pregnancy. Therefore, exercise specialists must inform and educate women about the benefits of exercise during pregnancy.

## 1. Introduction

According to recent evidence, regular physical exercises during pregnancy are associated with benefits for the body and mind and are safe for both mother and foetus [1,2].

The anatomical and physiological changes during pregnancy result in a variety of symptoms affecting the musculoskeletal system. Low back pain and pelvic pain during pregnancy are the most common complications of pregnancy [3]. The prevalence of urinary incontinence has been documented to be high both during and after pregnancy [4,5]. High incidence of oedema, pain and discomfort in lower limbs and unsteady gait have been identified as common lower extremity problems experienced during pregnancy [6]. All of these symptoms affect the physical activity levels and quality of life of pregnant women. Reduced physical activity weakens muscles, specifically in the abdomen, leading to an increased risk of diastasis rectus, which is also related to lumbopelvic instability and pelvic floor muscle weakness [7].

Different guidelines [8,9,10] recommend that pregnant women without contraindications should undertake physical activities regularly. Exercise interventions have also been shown to help prevent gestational diabetes [11] and preeclampsia [12] and facilitate and ease childbirth [13] or reduce the risk of caesarean section surgery [14]. Physically active pregnant women are less likely to become tired, have more energy, and their quality of life is shown to be better than that of physically inactive pregnant women [15]. Many pregnant women have limited knowledge about obesity, gestational weight gain and their consequences and management strategies [16].

Women who exercised before pregnancy and want to continue exercising during pregnancy despite physiological and psychosocial barriers need to receive information or social support regarding exercise [16]. Appropriate exercise during pregnancy may be beneficial in ensuring the safety of the pregnant women while promoting their physical and mental health [8]. No studies exploring the knowledge of the benefits of exercise and perceptions towards exercise during pregnancy in Lithuanian women were identified. The aim of this study was to explore women’s knowledge and perceptions of the effect of exercise during pregnancy.

## 2. Materials and Methods

*Setting*. This survey was conducted in Lithuania according to current ethical guidelines and was approved by the Institutional Review Board of Lithuanian Sports University.

*Sampling*. Women were eligible to participate in the study if they were able to read and understand Lithuanian language and were aged between 18–55 years. G*Power 3.1.9.2 test was used to calculate the required sample size. Test family: Chi-square tests. Effect size, default (0.3); Alpha err prob, default (0.05); Power (1-beta err prob), default (0.95); Total calculated sample size was 191.

*Recruitment*. Only women for this study were recruited using purposeful sampling for three groups with approximately the same number in each group. Thus, group 1 included pregnant women at the time of the study; group 2, women who were not pregnant but had given birth; and group 3, women who were not pregnant and had never given birth.

*Data collection*. The on-line questionnaire was distributed via a maternity-related social network (a web-based format using Web Surveyor, version 3.6; Web Surveyor Corporation, Herndon, VA). Replies were returned by 291 women.

*Questionnaire.* The questionnaire was developed by the researchers based on previous studies on knowledge and perceptions of exercise [17,18,19]. The designed questionnaire was content-validated by an expert’s reviews in a pilot study. Reliability and validity of the questionnaire tested was 0.9 and 0.89, respectively. The questionnaire consisted of 44 questions divided into two sections (knowledge and perceptions), and participants had to answer 15–20 questions, depending on which group they belonged to. First, nine questions were the same for all respondents and related to demographic information; then, the questions branched out depending on which group the respondent belonged to: (1) pregnant at the time of the study, (2) not pregnant but had given birth in the past, and (3) not pregnant and had never given birth. Women who were pregnant at the time of the study (group 1) had to answer extra questions about their health status, pregnancy trimester, and the number of pregnancies. Questions in the questionnaire were related to women’s physical activity levels and past and current health problems. Women who reported having health problems were directed to name those problems. The survey included open- and close-ended questions about respondents’ knowledge and perceptions of the effect of exercise during pregnancy. Completion of the questionnaire took up to 10 min.

All respondents were asked about their knowledge and perceptions of the role of exercise during pregnancy. Knowledge was classified as YES (I know), NO (I do not know), and NOT ENOUGH (I do not know enough). Perceptions, the understanding regarding the applicability of exercise in pregnancy, were classified as favourable or unfavourable, with opinions: YES, NO, and I do not know.

*Data analysis*. Descriptive statistics of mean and standard deviation and percentage were analysed for different variables related to knowledge and perceptions. Inferential statistics of the Chi square were used to determine any association between knowledge, perceptions, and respondent characteristics; *p* level was set at 0.05. Statistical Package for Social Science software version 24 for Windows was used.

## 3. Results

### 3.1. Sample Characteristics

The background characteristics of all study participants are shown in Table 1. The vast majority (78%; *n* = 227) of women in our study were young (aged 18–34 years).

### 3.2. Knowledge

More women who were not pregnant, but had given birth, had knowledge (*p* = 0.012) about exercise compared to the other two subgroups of women (Figure 1). However, in this group, almost half reported that they did not have enough knowledge.

Women who had knowledge about exercise (*n* = 163) were asked to answer the question about the sources of their knowledge (Figure 2).

The internet was the most common source of information for all three groups of women (Table 2). Respondent age did not have any influence (*p* > 0.05) on the sources of information as well as the group women belonged to.

### 3.3. Perceptions of the Effect of Exercise

The perceived aims of exercise reported by the women who had knowledge (*n* = 138) are listed in accordance with the number and % of answers (Table 3). More than half of the respondents thought that the purpose of exercise was to keep the body fit, reduce low back and pelvic pain, and facilitate childbirth.

The vast majority of respondents highlighted the importance of exercise in pregnancy and agreed with the statement that the obstetrician/gynaecologist should provide information about exercise classes during pregnancy (Table 4). A majority of the women (over 90%) replied positively to the questions about being active during pregnancy and receiving more information about the effect of exercise in pregnancy. Referral to physiotherapy for examination and advice was favoured by 91% of all respondents.

The need for information about exercises during pregnancy did not differ (*p* > 0.05) between the three groups of women (Table 5).

The results of this survey showed that the knowledge about exercises and perceptions of their effect on the body during pregnancy was, in general, at a very similar level among the three groups of women (*p* > 0.05).

## 4. Discussion

The current study showed that women who had given birth had more knowledge about exercise and its effect on the body in pregnancy compared to pregnant women or women who had never been pregnant. The vast majority of women believed that medical professionals should provide information about the effect of exercise in pregnancy. Even when the respondents did not have enough knowledge, they perceived the role of exercise positively.

Pregnancy represents a special phase in life when women are often highly motivated to implement behavioural changes. Although pregnant women are encouraged to participate in aerobic and strength training activities [20], they face a lack of knowledge on safe implementation of medical and scientific recommendations. Therefore, it is important to explore their knowledge of the effects of exercise on pregnancy, so that an appropriate and effective education and practice can be implemented in different phases of pregnancy. Health care professionals need to be aware of the level of knowledge of their clients and client groups, so that appropriate health promoting activities can be developed. When educating health care professionals, e.g., physiotherapists, a client-centred approach is one of the key factors influencing the development and planning of service. Therefore, it is important to learn and explore clients’ knowledge of the issues that are going to be implemented.

Women in our study were found to have positive perceptions towards exercise during pregnancy, and they perceived that exercises are effective in keeping the body fit during pregnancy. After analysing the study results, we found that only 24% of all women reported that they were aware of the impact exercise might have during pregnancy, while 32% reported they had some knowledge but not enough, and 44% of women said that they did not know anything about the effect of exercise for pregnancy. Nayak et al. [17] had similar results in their study, where 54% of women did not know anything about exercise in antenatal care. In the study by Mbada et al. [18], only pregnant women participated (*n* = 189), and a majority of them demonstrated inadequate knowledge of antenatal exercises; however, the women had positive attitudes towards exercise.

Women who were not pregnant but had given birth had significantly more knowledge about exercise compared to women who were pregnant at the time of the study and those who were not pregnant and had never given birth. Naturally, women who had never been pregnant or given birth had less knowledge about exercise during pregnancy. Surprisingly, though, more than 50% of the women who were pregnant at the time of the study had no knowledge about exercise for pregnancy, similarly to the percentage of women who had never been pregnant. Even when the respondents did not have enough knowledge, they perceived the role of exercise positively.

Childbirth is an important life experience for a woman [21]. In our study, 50% of women thought that exercise might help to facilitate childbirth; furthermore, a vast majority (96.4%) of all respondents believed they should be physically active during pregnancy and participate in activities such as Pilates, yoga and others. Women (91.8%) also reported the need for special courses, workshops, or educational materials. The vast majority (92%) of respondents (*n* = 138) who had knowledge about the effect of exercise believed that before planning pregnancy, they should physically prepare their body for the pregnancy. For their next pregnancy, most of the respondents (77.5%) reported that they were planning to exercise, and the answers to this question were not dependent on the current status of the women. The study comparing Australian (*n* = 215) and Chinese (*n* = 240) pregnant women’s knowledge about exercise found that Australian women reported higher levels of current exercise and intentions to exercise in the next 4 weeks of pregnancy compared with Chinese women. The authors concluded that beliefs, attitudes, barriers and intentions regarding exercise during pregnancy differ between cultures [19].

Our study also revealed that more than half of the respondents who had knowledge about the effect of exercise had found this information on the internet. In the age of digitization, the internet provides easy access to vast amounts of information on different topics [22] that is not always of the highest quality or reliable. Less than one-third of respondents had received this knowledge from a physiotherapist, and only 18.4% received information from their general practitioner. In the study by Nayak et al. [17], even fewer women (10%) received knowledge about the effect of exercise from health care professionals.

Since pelvic and low back pain are the most common musculoskeletal disorders experienced during pregnancy [23], the role of a physiotherapist might become essential in prescribing exercise and self-management strategies [24]. Only 13.7% of all women reported that they had knowledge about the applicability of physiotherapy in pregnancy, while 52.6% reported they did not know, and 33.7% of women said they knew, but not enough. Therefore, advice and support to pregnant women experiencing low back and pelvic pain should be individualized and evidence-based [25]. Self-management support should focus on behaviour change and an active lifestyle and should target attitude, self-efficacy, social influence, knowledge, and skills with regard to managing the condition [24].

Even though exercise may only have a small protective effect against low back pain in pregnancy [26], half of the women believed that exercise might be helpful. Respondents in our study believed that information about correct breathing, correct standing and sitting positions, body training, pelvic floor exercises and advice on adverse body positions would be useful.

Evidence shows that exercise during pregnancy is important to prevent or help with pregnancy-related complications [1], and it should be prescribed in clinical practice. Women who reported having knowledge about the effect of exercise during pregnancy were asked to clarify indications for exercise. The main reason given for consideration was body strengthening, and the less important reasons were low back and pelvic pain alleviation and facilitation of delivery. There are limitations in this research, and the main one concerns the sample selection. The sample was based on convenience sampling, and study was conducted on a sample of Lithuanian women. The results may not be generalizable across different nations. However, this was the first study to explore women’s knowledge and perceptions of the effect of exercise for pregnancy in Lithuania. The sample size (291) included 0.02% of all Lithuanian women. To obtain a more in-depth account of women’s knowledge, perceptions, and experiences of exercise during pregnancy, qualitative research is needed. It would also be beneficial to explore health professionals’ knowledge and perceptions of the topic so that they would be able to advise pregnant women to seek the support of exercise and movement specialists at a special time in their lives. The researchers suggest applying exercise programs that would be supervised by exercise specialists, preferably qualified according to international educational standards [27]. In summary, the findings of this study provide a useful foundation for the implementation of education for pregnant women, which is key to improving antenatal care [28]. The results may be of interest to general practitioners, gynaecologists, midwives, physiotherapists, exercise specialists, and other health professionals who are engaged in the care of pregnant women.

## 5. Conclusions

Knowledge about the effect of exercise during pregnancy was found to be low. Almost all women who responded to the survey expressed a need for knowledge about the effect of exercise. This demands that exercise specialists and other healthcare professionals encourage women to be active and educate them about potential benefits and forms of exercise during pregnancy and postpartum.

## Figures and Tables

**Figure 1 ijerph-20-01822-f001:**
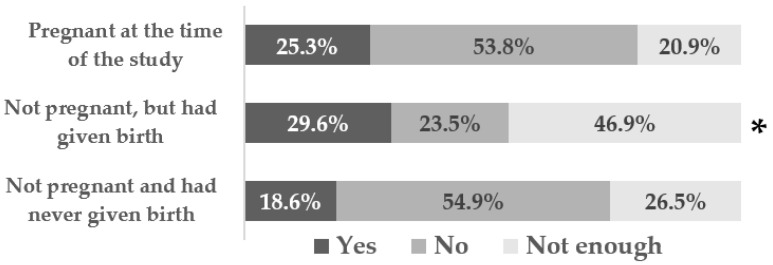
Distribution of answers to the question “Do you have knowledge about exercise for pregnancy” according to the current status of the respondents; *—*p* < 0.05 comparing first group with second, and second group with third.

**Figure 2 ijerph-20-01822-f002:**
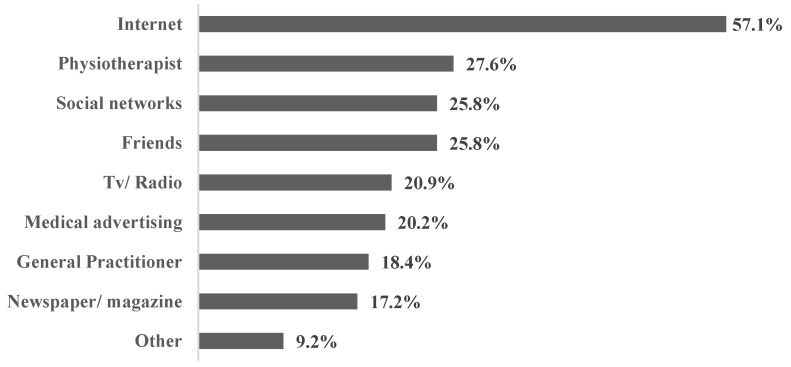
Distribution of the answers to the question “Where did you get information about exercise for pregnancy”.

**Table 1 ijerph-20-01822-t001:** Characteristics of the study participants.

Participants	% (Frequency)
Age (years)	
18–24	39.9 (116)
25–34	38.1 (111)
35–55	22 (64)
State of women	
Not pregnant but had given birth in the past	33.3 (97)
Not pregnant and had never given birth	35.4 (103)
Pregnant at the time of the study	31.3 (91)
Pregnancy trimester	
First trimester	36.3 (33)
Second trimester	47.3 (43)
Third trimester	16.4 (15)
Number of pregnancies	
First pregnancy	71.4 (65)
Second pregnancy	16.5 (15)
Third pregnancy	7.7 (7)
Four and more pregnancies	4.4 (4)

**Table 2 ijerph-20-01822-t002:** Distribution of the answers to the question "Where did you find out about exercise for pregnancy?" depending on respondents’ status.

Sources of Information	Pregnant at the Time of Study, % (*n*)	Not Pregnant, but Have Given Birth, % (*n*)	Not Pregnant and Have Never Given Birth, % (*n*)
Internet	30.5 (28)	29.4 (29)	26.1 (27)
Social networks	18.3 (17)	9.2 (9)	14.8 (15)
Television/radio	3.7 (3)	14.4 (14)	10.2 (11)
Newspaper/magazine/booklets	3.7 (3)	11.1 (11)	9.1 (9)
Friends	8.5 (8)	13.1 (13)	17 (18)
General practitioner/doctor	8.5 (8)	11.8 (11)	5.7 (6)
Physiotherapist	22 (20)	9.2 (9)	13.6 (14)
Other	4.9 (4)	2 (2)	3.4 (4)

**Table 3 ijerph-20-01822-t003:** Distribution of all answers to the question "Why should women attend exercise classes during pregnancy?".

Variables	% (*n*)
To keep body fit	58 (80)
To reduce low back and pelvic pain	55.1 (76)
To facilitate childbirth	50.7 (70)
To reduce stress	47.1 (65)
To better supply the foetus with oxygen	40.6 (56)
To learn specific exercises	33.3 (46)
To reduce swelling in the lower extremities	31.2 (43)
To assure the normal course of pregnancy	29.7 (41)
To increase elasticity of the perineum	26.1 (36)
To keep normal muscle tone	25.4 (35)
To assure normal foetal position in the uterus	16.7 (23)
To prevent muscle spasms	15.9 (22)
To reduce labour pain	9.4 (13)
To prevent high blood pressure	6.5 (9)
To prevent eclampsia	0.7 (1)
To prevent preeclampsia	0.7 (1)
Do not know	11.6 (16)

**Table 4 ijerph-20-01822-t004:** Distribution of answers regarding the importance of and information on exercise in pregnancy.

Question	Answers % (*n*)		
	Yes	No	Do Not Know
Do you think that when planning pregnancy, you should prepare your body for it?	92 (127)	2.2 (3)	5.8 (8)
Should a woman be physically active (Pilates, yoga, other activities) during pregnancy?	96.4 (133)	0 (0)	3.6 (5)
Do you plan to exercise during the first/next pregnancy?	77.5 (107)	5.1 (7)	17.4 (24)
Should the obstetrician/gynaecologist provide information about exercise during pregnancy?	88 (256)	1 (3)	11 (32)
Should pregnant women be referred to a physiotherapist for an examination and advice?	90.7 (264)	0.7 (2)	8.6 (25)
Are special courses/workshops/leaflets and other materials related to therapeutic exercise needed during pregnancy?	91.8 (267)	0.7 (2)	7.6 (22)
Would you recommend pregnant women to attend exercise classes?	62.5 (182)	0.7 (2)	36.8 (107)

**Table 5 ijerph-20-01822-t005:** Distribution of answers to the question "What information about exercise and its effect during pregnancy you would like to know".

Variables	Pregnant at the Moment, % (*n*)	Not Pregnant, but Have Given Birth, % (*n*)	Not Pregnant and Have Never Given Birth, % (*n*)
Correct breathing	13.2 (12)	15.5 (15)	14.6 (15)
Correct standing and sitting position	11.0 (10)	10.3 (10)	14.6 (15)
Correct body training	17.6 (16)	18.6 (18)	15.5 (16)
Correct stretching exercises	11.0 (10)	11.3 (11)	12.6 (13)
Exercises and body positions to be avoided	20.9 (19)	18.6 (18)	17.5 (18)
Correct pelvic floor exercises	18.7 (17)	10.3 (10)	10.7 (11)
Kegel exercises	6.6 (6)	9.3 (9)	10.7 (11)
Do not know	1.1 (1)	6.2 (6)	3.9 (4)

## Data Availability

Not applicable.

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
