# Peer review of "Women’s Knowledge and Perceptions of the Effect of Exercise during Pregnancy: A Cross-Sectional Study"

_ijerph, 2023, doi:10.3390/ijerph20031822_

Round 1

Reviewer 1 Report

Please see word document attached. 

Author Response

Response to Reviewer 1

We thank the reviewer for the comments and questions, and hope that we improved our article after making corrections. We tried to pay attention to all comments and remarks.

Abstract:

Line 9, Lines 13-16, Line 17, Lines 16-18, Line 21 – we agree with the remarks and corrected Abstract.

Introduction:

Lines 24-25 – English language specialist left this sentence without corrections.

Line 38 – we agree with the remark, we made mistake citing this sentence.

Line 45 – we corrected aim of the study.

Methods:

Line 53, Line 56 we made corrections.

Lines 62-63: There have been done studies with pregnant women in other populations. This is the first study in Lithuania and to our knowledge in general comparing three subgroups of women. We didn’t manage to find similar study.

Results:

Table 1 – after discussion with the statistician we did not make corrections in this table.

Figure 2 and Table 2 - after discussion with the statistician we did not make corrections in this table. Figure 2represents results of all study participants, and Table 2 – of three subgroups of women, and these results did not duplicate each other.

Line 107, Line 108, Lines 112-113, Lines 118-119 – we made corrections in these lines. But still one question was related to knowledge about physiotherapy, not only about therapeutic exercise and their effect.

Tables 1-5 – statistical difference was found only in Figure 1, - we corrected Figure 1. There was no statistical difference between groups in other results, and this is mentioned in the text.

Discussion:

Line 129, Line 138, Lines 139-142, Line 144, Line 146, Line 150, Line 158,  - we made corrections to these remarks.

Line 160 - one question was related to knowledge about physiotherapy, not only about therapeutic exercise and their effect. That is why we use this term here.

Line 163, Line 164, Lines 163-165, Line 172, – we made corrections to these remarks.

Line 174 indications to exercise means conditions, when exercises may be applied.

Line 185, Lines 186-189, Lines 190-195 – we made corrections and answered to these comments.

Reviewer 2 Report

Thank you for the opportunity to review this work. The authors have explored women’s knowledge and perceptions about therapeutic exercise (TE) during pregnancy. They found that overall knowledge of TE was low. It was nice that the authors focused on knowledge related to exercise during pregnancy. Exercise can have beneficial effects during pregnancy, and it is a good area to examine.

It would have been interesting if the authors would have considered a mixed methods approach for this study by including a qualitative method in their study. However, they do recommend it as an option for future studies.

Specific Comments

·       The introduction should include a little more about the population under study. In general, the introduction can be expanded and provide data if they exist on TE during pregnancy.

·       There needs to be a rationale on why women living in Lithuania were selected. How does this population differ from others? The authors should also mention the generalizability of the findings in the discussion.

·       Why were the age groups expanded to include women beyond 45 years?

·       How was sample size determined? Was a power calculation done?

·       Were the questionnaires developed for this study pre-tested? How were reliability and validity of the questionnaires tested?

·       The results need to be presented in terms of statistical tests done to measure association. All the tables are representing n (%). The authors mention in the methods that they will be performing a Chi-square test? Why have these associations not been presented? Additional analyses would really strengthen this paper.

·       The beginning of the discussion seems to re-state the objectives which is not necessary. The authors should only provide a summary of findings.

·       The first two paragraphs of the discussion need to be reworded. The numbers already presented in the results should not be repeated. Some parts of the discussion seems to mention what is already presented in the results.

·       Please list some limitations of the study.

·       Have similar studies been done in other populations?

Author Response

We thank the reviewer for the comments and questions, and hope that we improved our article after making corrections. We tried to pay attention to all comments and remarks. (Q – question; R – response).

Q. The introduction should include a little more about the population under study. In general, the introduction can be expanded and provide data if they exist on TE during pregnancy.

R. We expanded a little bit this section.

Q. There needs to be a rationale on why women living in Lithuania were selected. How does this population differ from others? The authors should also mention the generalizability of the findings in the discussion.

R. Women living in Lithuania were chosen because the knowledge and opinions of Lithuanian pregnant women have never been investigated before, and the main researcher is Lithuanian. We mentioned about the generalizability.

Q. Why were the age groups expanded to include women beyond 45 years?

R. Because one of subgroups were women not pregnant and had never given birth. Besides nowadays there is a tendency to give birth in older age.

Q. How was sample size determined? Was a power calculation done?

R. G*Power 3.1.9.2 test was used for sample size assessing. Test family: Chi-square tests. Statistical test: Goodness-of-fit tests: Contingency tables. Type of power analysis: A priori: Compute required sample size – given alpha, power, and effect size. Input parameters: Effect size, default (0.3); Alpha err prob, default (0.05); Power (1-beta err prob), default (0.95); Total sample size calculated - 191.

Q. Were the questionnaires developed for this study pre-tested? How were reliability and validity of the questionnaires tested?

R. The designed questionnaire was content validated by expert’s reviews in a pilot study. Reliability and validity of the questionnaires tested were 0.9 and 0.89 respectively.

Q. The results need to be presented in terms of statistical tests done to measure association. All the tables are representing n (%). The authors mention in the methods that they will be performing a Chi-square test? Why have these associations not been presented? Additional analyses would really strengthen this paper.

R. The Chi-square test result is presented in the text and also, we included it into the Figure 1. Also, we mentioned in the text that there was no statistical difference between the three subgroups describing tables.

Q, The beginning of the discussion seems to re-state the objectives which is not necessary. The authors should only provide a summary of findings.

R. We corrected this paragraph.

Q. The first two paragraphs of the discussion need to be reworded. The numbers already presented in the results should not be repeated. Some parts of the discussion seems to mention what is already presented in the results.

R. We corrected these two paragraphs.

Q. Please list some limitations of the study.

R. We listed limitations in the Discussion section.

Q. Have similar studies been done in other populations?

R. Yes, similar studies are done in Pakistan, India, Africa, UAE.

Round 2

Reviewer 1 Report

Please see the document attached. 

Author Response

We thank the reviewers for pointing out the inaccuracies in the manuscript. Those comments are all valuable and very helpful for revising and improving our paper.

To avoid confusions between “therapeutic exercises” and “the effect of TE”, we adjusted the use of terms. We have decided to use the term “exercise” and “the effect of exercise” throughout the text, removing the word “therapeutic”, because in the introduction we are talking about the effects of exercise and physical activity in general. And even in the questionnaire we ask about Yoga and Pilates – as forms of activities.

Abstract

  1. We corrected all remarks and use in the text word “exercise” and benefits or effects of exercise.

Introduction:

  1. We corrected this section avoiding term Therapeutic.

Methods:

  1. We corrected this section avoiding term Therapeutic.

Results:

  1. We corrected table 1 and Figure 1. And we corrected line 149 including words: knowledge about exercise and their effect. We moved (Table 4) to end of the sentence. We included statistical data in this section.

Discussion:

  1. We corrected this section by discussing more studies and highlighted the implications and importance by adding text: Therefore, it is important to explore their knowledge on the effects of exercise on pregnancy, so that an appropriate and effective education and practice can be implemented in different phases of pregnancy. Health care professionals need to be aware of the level of knowledge of their clients and client groups, so that appropriate health promoting activities can be developed. When educating health care professionals, e.g., physiotherapists, client centred approach is one of the key factors influencing the development and planning of service. Therefore, it is important to learn and explore clients’ knowledge on the issues that are going to be implemented.

Reviewer 2 Report

Minor comments:

Line 56: please either state what the three groups refer to or you can remove this word from the introduction because it is a part of the methods.

Line 66: List the power along with sample size details.

Line 75: Somewhere in the methods please mention the pretesting of the questionnaire along with the reliability and validity.

Discussion: please begin the discussion with a summary of findings. The objectives need not be restated in the beginning of the discussion.

Author Response

We thank the reviewers for pointing out the inaccuracies in the manuscript. Those comments are all valuable and very helpful for revising and improving our paper.

Response to comments:

Line 56: please either state what the three groups refer to or you can remove this word from the introduction because it is a part of the methods.

We corrected this sentence by removing three groups.

Line 66: List the power along with sample size details.

We included sample size details.

Line 75: Somewhere in the methods please mention the pretesting of the questionnaire along with the reliability and validity.

We included this in lines 82-83

Discussion: please begin the discussion with a summary of findings. The objectives need not be restated in the beginning of the discussion.

We corrected this section.
